Revalidation of morphological characteristics and multiplex PCR for the identification of three congener invasive Liriomyza species (Diptera: Agromyzidae) in China

Chang Ya-Wen 1
Chen Jing-Yun 1 2
Zheng Si-Zhu 2
Gao Yuan 2
Chen Yunfang 2
Deng Yanfeng 2
Du Yu-Zhou yzdu@yzu.edu.cn 1 3
1 College of Horticulture and Plant Protection & Institute of Applied Entomology, Yangzhou University , Yangzhou , China
2 Suzhou Customs , Suzhou , China
3 Joint International Research Laboratory of Agriculture and Agri-Product Safety, Yangzhou University , Yangzhou , China
Gillespie Joseph
Electronic publication date: 2020 Oct 30
Publication date: 2020
Volume: 8
Electronic Location ID: e10138
Received 2020 Jun 26; Accepted 2020 Sep 18
Copyright: ©2020 Chang et al.
Copyright year: 2020
Copyright holder: Chang et al.
License: This is an open access article distributed under the terms of the Creative Commons Attribution License, which permits unrestricted use, distribution, reproduction and adaptation in any medium and for any purpose provided that it is properly attributed. For attribution, the original author(s), title, publication source (PeerJ) and either DOI or URL of the article must be cited.
License URL: https://creativecommons.org/licenses/by/4.0/

Keywords: Liriomyza, Morphological characteristics, Abdominal tergites, Multiplex PCR, Species identification, COI

Funding: Suzhou Customs Science and Technology Program 2020SZKY05 Agricultural Industry Technology System JATS [2019] 331 Jiangsu Science and Technology Support Program BE2014410 The Basic Research Program of Agricultural Application of Suzhou SNG201602 This research was funded by the Suzhou Customs Science and Technology Program (2020SZKY05), the earmarked fund for Jiangsu Agricultural Industry Technology System (JATS [2019] 331), the Jiangsu Science and Technology Support Program (BE2014410), and the Basic Research Program of Agricultural Application of Suzhou (SNG201602). The funders had no role in study design, data collection and analysis, decision to publish, or preparation of the manuscript.

==============================
Due to varietal differences, diminutive size, and similar morphological characters, it is difficult to classify and identify Liriomyza spp., a genus comprised of economically-important, highly-polyphagous insect pests. In this study, we reconfirmed the morphological characteristics of three closely-related invasive leafminers, L. trifolii, L. sativae, and L. huidobrensis. Morphological results showed that characteristics imparted by the male genitalia were the most reliable morphological features for identification. The colors exhibited by vertical setae were variable among species, and the ratio of the length of the ultimate section of vein CuA1 divided by penultimate section also varied within species. Although the patterns of abdominal tergites were diverse among Liriomyza spp., L. trifolii exhibited a unique pattern with a yellow patch at the 5th black visible tergite; this pattern can be profiled as a prominent characteristic for morphological identification. In order to identify the three Liriomyza spp. quickly and accurately, we developed an improved molecular identification method using multiplex PCR based on the gene encoding mitochondrial cytochrome oxidase I (COI); this method enabled direct identification based on the size of amplified products. The results of this study provide a valuable reference for the identification of Liriomyza spp., which will ultimately improve our ability to control individual species.

Introduction

Leafminer flies (Diptera: Agromyzidae), especially Liriomyza trifolii, L. sativae and L. huidobrensis, are invasive insect pests in many countries. They are polyphagous, economically-significant pests that cause severe damage to many ornamental and vegetable crops worldwide (Spencer, 1973; Spencer, 1990; Reitz et al., 1999). Both larvae and adults cause serious damage to crops (Musgrave, Poe & Bennett, 1975; Minkenberg & Van Lenteren, 1986). The damage caused by larval feeding on leaves can reduce photosynthetic capacity, and leaf mining activity can cause premature leaf drop resulting in reduced yields (Johnson et al., 1983; Chandler & Gilstrap, 1987). Moreover, indirect damage occurs when adults pierce leaves for feeding and oviposition, thus increasing plant susceptibility to disease (Zitter & Tsai, 1977; Motteoni & Broadbent, 1988). The rapid life cycle and high growth rate of Liriomyza spp. can lead to serious crop losses. Accurate identification of Liriomyza is important for implementing effective control strategies, because insecticide resistance and tolerance to environmental stress varies among species (Chang et al., 2017; Gao et al., 2017).

Closely-related Liriomyza spp. are similar in morphology at the adult stage (Oudman et al., 1995; Lei, Wang & Wen, 1996; Chen, 1999; Scheffer et al., 2001), and adult males can only be identified with certainty according to genitalia, which is both time-consuming and difficult. Identification at the early developmental stages of Liriomyza infestation is necessary for effective control; however, the absence of morphological characters makes identification difficult and larvae cannot be collected directly due to their mining behavior (Oudman et al., 1995; Chiu et al., 2000; Morgan et al., 2000; Scheffer et al., 2001).

Since morphological identification of female adults, larvae and pupae of Liriomyza species is complex and difficult, molecular methods of identification are required. Immature developmental stages are the most common forms intercepted at ports of entry, therefore, it is important to identify these interceptions accurately and rapidly. With the development of mitochondrial and other molecular markers (Carapelli et al., 2018; Chen et al., 2019), several molecular methods have been developed to identify Liriomyza species (Menken & Ulenberg, 1983; Zehnder, Trumble & White, 1983; Oudman et al., 1995; Chiu et al., 2000; Morgan et al., 2000). Multiplex PCR is a cost-effective, rapid, accurate method where identification can be determined by PCR product size with species-specific primers (Nakamura et al., 2013).

In this study, we re-verified morphological characteristics of three leafminers, L. trifolii, L. sativae and L. huidobrensis. A new morphological characteristic for detection of L. trifolii was investigated, and an improved molecular method for identification was developed based on multiplex PCR. This study provides approaches that can be deployed for identification of Liriomyza species, which will ultimately help future control efforts.

Materials and Methods

Insects

The three species of Liriomyza spp. were collected from areas where leafminers occur in China. In this study, 263 individuals of three species were selected for further data analysis (Table S1). These were collected at the larval stage, tagged with relevant information and transported to the laboratory for pupation and emergence as adults. After preliminary morphological identification, adults were labeled, immersed in 70% ethanol and stored at −20 °C. After dissecting and photographing the samples, the remaining tissues were stored in 100% ethanol for DNA extraction and molecular analysis.

Morphological identification

Samples were examined with a stereomicroscope (Zeiss Stemi 2000c) and photographed with a wide depth of field (Zeiss Smartzoom 5). Male genitalia and wings were dissected, and slides were prepared and photographed with the Axio imager A2 (Zeiss, Germany).

Differences in the ratios of ultimate section lengths of vein CuA1 among different Liriomyza species were determined by one-way analysis of variance (ANOVA), followed by Tukey’s multiple comparisons. All statistical analyses were performed using SPSS v. 16.0 (SPSS, Chicago, IL, USA), and statistical significance was determined when P <  0.05.

Molecular identification and primer selection for multiplex PCR

Genomic DNA of Liriomyza species was extracted using the AxyPrep™ Multisource Genomic DNA Kit (Axygen, USA). A partial sequence of the mitochondrial cytochrome oxidase I (COI) gene was amplified with common primers F, 5′-CAACATTTATTTTGATTTTTTGG-3′ and R, 5′- TCCAATGCACTAATCTGCCATATTA-3′ (Simon et al., 1994; Yang, Cao & Du, 2010) using protocols described by Chen et al. (2019), to molecular cross-checking and verification all of Liriomyza species in this study using sequencing, accession number can be found in Table S1.

For multiplex PCR, full-length COI genes of three Liriomyza species were downloaded from NCBI (https://www.ncbi.nlm.nih.gov/) and aligned using Clustal X. To develop a rapid identification method, three species-specific primers and a common reverse primer were mixed to amplify DNA from different Liriomyza species. The PCR conditions were as follows: denaturation at 94 °C for 3 min; 35 cycles at 94 °C for 1 min, 58 °C for 1 min and 72 °C for 1 min; followed by extension at 72 °C for 10 min. PCR was conducted in a 25 µL reaction volume containing 2 µL (100 ng) of DNA template, 1 µL (10 µM) of each primer, 12.5 µL of 2 × Taq Master mix (Vazyme Biotech Co., Ltd) and 6.5 µL ddH2O. PCR products were separated in 1.0% agarose gels, and primers that amplified only one specific band for each species are shown in Table 1.

Table 1 Information of the primers designed in this study.

Primer name	Nucleotide sequence (5′–3′)	Ta (Tm) °C	Product size (bp)	GenBank number	
Lt612	CAATTACAATACTATTAACAGACCG	58 (48.5)	569	MT919718	
Ls262	AGCTCCAGACATAGCATTTCCTCG	58 (58.9)	919	MT919719	
Lh959	TTCAGATGGCTTGCCACATTACACG	58 (59.9)	222	MT919720	
LR1181	GAATAAATCCKGCTATAATTGCAAATAC	58 (50.9)	–	–	

Results

Morphological identification

The distiphallus, which is part of the male genitalia, is a very small, fragile structure enclosed by membranes located at the terminus of the aedeagus. For L. trifolii, the morphological characteristics of the distiphallus include one distal bulb with marked constriction between lower and upper halves in dorsoventral view; the bulb is lightly sclerotized with a long basal stem (Fig. 1A). For L. sativae, the distiphallus is characterized by one distal bulb with a slight constriction between upper and lower halves in the dorsoventral view; the bulb is more intensely sclerotized with a shorter basal stem (Fig. 1B). For L. huidobrensis, the distiphallus contains two distal bulbs; these meet at rims that extend in an anteroventral orientation (Fig. 1C).

Figure 1 Photo plates of the phalluses of three Liriomyza species, lateral view.

(A) L. trifolii; (B) L. sativae; C, L. huidobrensis. Arrows indicate the distiphallus. Scale bar = 0.01 mm.

With respect to vertical setae, L. trifolii exhibits inner and outer vertical setae on a yellow background; whereas vertical setae are present on a black background for L. huidobrensis. In L. sativae, outer and inner vertical setae are presented on black and yellow backgrounds, respectively (Spencer, 1973). In this study, only 86.1% (192/223) of L. trifolii had yellow inner and outer vertical setae; 9.9% (22/223) had yellow inner vertical setae and undetermined color for outer setae, and 4.0% (9/223) had yellow inner and black outer vertical setae (Table 2; Figs. 2A–2C). For L. sativae, 17.6% (6/34) had black inner and outer vertical setae, 58.8% (20/34) had yellow inner and black outer vertical setae, and 23.5% (8/34) had outer black setae with an undetermined color for inner vertical setae (Table 2; Figs. 2D–2F). For L. huidobrensis, 100% (6/6) exhibited black inner and outer vertical setae (Table 2; Figs. 2G–2I). These results show that characteristics of vertical setae are not reliable for identifying Liriomyza species.

Table 2 The data of color characteristics of outer and inner vertical setae in three Liriomyza species.

Species	Vertical setae position (Inner/Outer)	Individual phenotypes	Vertical setae position (Inner/Outer)	Individual phenotypes	Vertical setae position (Inner/Outer)	Individual phenotypes	
L. trifolii	Y/Y	192	B/Y	0	U/Y	0	
Y/B	9	B/B	0	U/B	0	
Y/U	22	B/U	0	U/U	0	
L. sativae	Y/Y	0	B/Y	0	U/Y	0	
Y/B	20	B/B	6	U/B	8	
Y/U	0	B/U	0	U/U	0	
L. huidobrensis	Y/Y	0	B/Y	0	U/Y	0	
Y/B	0	B/B	6	U/B	0	
Y/U	0	B/U	0	U/U	0	
Notes.

Abbreviations Y yellow

B black

U unclear

Figure 2 The color characteristic of outer and inner vertical setae position in three Liriomyza species.

(A–C) L. trifolii; (D–F), L. sativae; G-I, L. huidobrensis. Scale bar=0.1 mm. The yellow arrow indicated the position of outer vertical setae and the red arrow indicated the position of inner vertical setae.

Wing pattern ratios were calculated as the length of the ultimate section of vein CuA1 divided by the penultimate section (‘a’ and ‘b’, see Figs. 3A–3C). In this study, ‘a’ was 2.70 ±  0.31 times the length of ‘b’ in L. trifolii, and ‘a’ was 2.72 ±  0.37 times the length of ‘b’ in L. sativae. For L. huidobrensis, ‘a’ was 2.20 ± 0.24 times the length of ‘b’ (F2,237 = 7.345, P <  0.05) (Fig. 4). Although the ratio of L. huidobrensis was significantly different from the other two species (P <  0.05), there was no significant difference between L. trifolii and L. sativae ( P = 0.907). Many L. trifolii individuals exhibited truncated or missing dm-cu cross veins. Furthermore, we noted inconsistency between left and right forewing patterns within individual samples (Fig. 3A, with dashed lines).

Figure 3 Comparison of the wing patterns of three Liriomyza species. The length of ultimate section of vein CuA_1 divided by penultimate section (a and b sections).

(A), L. trifolii; (B), L. sativae; (C), L. huidobrensis. Scale bar=0.1 mm. Dot box represents abnormal wing pattern in L. trifolii.

Figure 4 The ratio of the length of ultimate section of vein CuA1 divided by penultimate section.

Differences in the ratio length of ultimate section of vein CuA1 among different Liriomyza species were determined by one-way analysis of variance (ANOVA), followed by Tukey’s multiple comparison (P < 0.05). The data in the figure is the average ± standard deviation.

In L. trifolii, the 2nd–5th visible tergites were generally divided by a yellow medial furrow in male adults; furthermore, there was a yellow patch at the 5th black visible tergite that can distinguish L. trifolii from other Liriomyza species (Figs. 5A–5C). In L. sativae and L. huidobrensis, only the second visible tergite is divided by a yellow medial furrow and no yellow patch is evident on the 5th tergite (Figs. 5D–5I).

Figure 5 Diagrams of abdominal color patterns of three Liriomyza species.

(A–C) L. trifolii; (D–F), L. sativae; (G–I), L. huidobrensis. Scale bar = 0.1 mm.

Molecular detection of Liriomyza spp.

Candidate primers for species-specific detection of Liriomyza were based on the alignment of 262 (L. sativae), 612 (L. trifolii), and 959 (L. huidobrensis) COI sequences. We designed one reverse primer, 1181 R, that was common to all three Liriomyza species. The position of forward primers was selected to produce < 1,000 bp amplicons when paired with the reverse primer with at least 300 bp nucleotides between species. In addition, sites were selected where the number of differential nucleotides was >2 bp to increase the specificity of the primers (Fig. 6).

Figure 6 Alignment of COI sequences. Boxs indicate primer positions used in this paper.

Base substitutions are indicated by the shadow. Lt, L. trifolii; Ls, L. sativae; Lh, L. huidobrensis.

The three Liriomyza species could be differentiated by specific PCR products in 1.0% agarose gels, and the resulting PCR products were 569, 919, and 222 bp for L. trifolii, L. sativae and L. huidobrensis, respectively (Fig. 7A). The validity of multiplex PCR for identification was further confirmed by using the system with different developmental stages; the approach worked equally well for larvae, pupae and adults of the three Liriomyza species (Fig. 7B). Populations from different geographical regions were also obtained to evaluate the reliability of species-specific primers. The results obtained by multiplex PCR (Fig. S1) and subsequent sequence analysis of COI (Fig. S2) showed that geography did not impact the reliability of primers.

Figure 7 Agarose gel electrophoresis image of multiplex PCR products.

(A) DNA from different Liriomyza adults. (B) DNA from different developmental stages of L. trifolii. Each experiment has three biological repeats. Lt, L. trifolii; Ls, L. sativae; Lh, L. huidobrensis.

Discussion

The morphological characteristics used for Liriomyza identification have primarily followed Spencer’s (1973) criteria. However, variability in life stages, emergence times and sample preservation result in large differences in body color and markings, which can make current morphological criteria unreliable for identification (Spencer, 1973; Kang, 1996; Shiao, 2004).

Currently, the identification of Liriomyza spp. based on morphology is restricted to male adults because there are no reliable features for species-level identification of female adults or immature developmental stages (EPPO, 2005). The identification of adults requires the examination of the male adult genitalia. In general, the distiphallus provides reliable detection of the three Liriomyza species and has considerable diagnostic value (Spencer, 1973; Shiao, 2004). However, differences in distiphalluses between species are subtle and dissection is difficult for nonprofessionals. Consequently, features of distiphallic structure should be cross-checked with other external morphological characteristics to ensure that identification is valid.

According to Spencer (1973), coloration of the vertical setae is an important external feature that can distinguish L. trifolii and L. sativae without dissection; however, this feature is unstable and lacks clear interspecific boundaries. Results of the current study show that reliance on coloration of vertical setae can result in misidentification of L. trifolii and L. sativae; thus, this feature should only be used as a supplement for identification. The ratio of the length of the ultimate section of vein CuA1 is unreliable since most ratio values overlapped among Liriomyza species. In this study, we also evaluated the patterns of abdominal tergites and discovered that the yellow patch at the 5th black visible tergite of L. trifolii is a new, reliable morphological characteristic for identification. Similar findings were reported for abdominal color patterns for six Liriomyza species (Shiao, 2004).

Molecular methods for insect identification can be used with different developmental stages, including immature stages where morphological features may be lacking. Furthermore, molecular assays may facilitate identification of atypical or damaged samples. However, the specificity of molecular assays may be limited because they were developed for a particular purpose and evaluated against a restricted number of species (Nakamura et al., 2013). Multiplex PCR assays were recently developed for identification of Liriomyza species (Miura et al., 2004; Guan et al., 2006; Nakamura et al., 2013) and are based on amplification of a target gene region using species-specific primer combinations. Multiplex PCR assays are easier and faster than other molecular methods, such as RAPD-PCR, PCR-RFLP, DNA barcoding and real-time PCR (Chiu et al., 2000; Morgan et al., 2000; Scheffer et al., 2001; Kox et al., 2005; Scheffer, Lewis & Ravindra, 2006; Blacket et al., 2015; Sooda et al., 2017); furthermore, multiplex PCR assays are more sensitive than enzyme electrophoresis methods (Zehnder, Trumble & White, 1983; Menken & Ulenberg, 1983; Minkenberg & Van Lenteren, 1986; Oudman et al., 1995). In general, the reliability and sensitivity of multiplex PCR represents a great improvement in molecular identification protocols and will enable us to manage invasive pests more effectively.

Conclusions

Invasive Liriomyza spp. comprise a group of insect pests that cause considerable economic loss and serious quarantine problems. In this study, morphological features were re-evaluated for L. trifolii, L. sativae, and L. huidobrensis, and the discriminative ability of traditional morphological characteristics, such as male genitalia, abdominal color patterns, length of CuA1 and abdominal tergite patterns were reevaluated. Furthermore, we developed an improved molecular identification method using multiplex PCR based on COI to identify the three Liriomyza species quickly and accurately. This study provides valuable tools for the identification of Liriomyza spp. using both morphological and molecular criteria.

Supplemental Information

Supplemental Information 1 Raw data: Ratio of the length of the vein CuA1 ultimate section divided by the penultimate section

Click here for additional data file.

Supplemental Information 2 Agarose gel electrophoresis of multiplex PCR products from different geographical populations of Liriomyza

Lt Lanes 1-10 indicate L. trifolii populations from: (1) Hengshui, (2) Hangzhou, (3) Dongguan, (4) Zhangzhou, (5) Qionghai, (6), Nanning, (7) Changzhou, (8) Nanchang, (9), Huizhou, and (10) Huzhou. Ls lanes 1-2 indicate L. sativae populations from Shangqiu and Luoyang, respectively. Lh lanes 1-2 represent L. huidobrensis populations from Kunming and the laboratory, respectively.

Click here for additional data file.

Supplemental Information 3 Alignment of COI sequences from different geographical population using species-specific primers

The shaded nucleotides represent divergent sites. Nucleotides bounded by red and blue dashed rectangles represent forward and reverse primers, respectively. Lt sequences represent the following L. trifolii populations: Lt 1, Hengshui; Lt 2, Hangzhou; Lt 3, Dongguan; Lt 4, Zhangzhou; Lt 5, Qionghai; Lt 6, Nanning; Lt 7, Changzhou; Lt 8, Nanchang; Lt 9, Huizhou; and Lt 10, Huzhou. Ls sequences represent L. sativae populations from Shangqiu (Ls 1) and Luoyang (Ls 2). Lh sequences represent L. huidobrensis populations from Kunming (Lh 1) and the laboratory (Lh 2).

Click here for additional data file.

Supplemental Information 4 List of sample collection information

Click here for additional data file.

Additional Information and Declarations

Competing Interests

Author Contributions

DNA Deposition

Data Availability

Jing-Yun Chen, Si-Zhu Zheng, Yuan Gao, Yun-Fang Chen, and Yan-Feng Deng are employed by Suzhou Customs.

Ya-Wen Chang and Jing-Yun Chen conceived and designed the experiments, performed the experiments, analyzed the data, prepared figures and/or tables, authored or reviewed drafts of the paper, and approved the final draft.

Si-Zhu Zheng, Yuan Gao, Yunfang Chen and Yanfeng Deng analyzed the data, prepared figures and/or tables, and approved the final draft.

Yu-Zhou Du conceived and designed the experiments, authored or reviewed drafts of the paper, and approved the final draft.

The following information was supplied regarding the deposition of DNA sequences:

The COI sequences amplified by pecies-specific primer designed in this study are available at GenBank: MT919718 (L. trifolii), MT919719 (L. sativae) and MT919720 (L. huidobrensis).

Data used for molecular cross-checking and verification all of Liriomyza species in this study were MT932588–MT932810 (L. trifolii), MT926413–MT926446 (L. sativae), and MT926447–MT926452 (L. huidobrensis).

The following information was supplied regarding data availability:

The raw measurements are available in the Supplemental Files.

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
