# Peer review of "Revalidation of morphological characteristics and multiplex PCR for the identification of three congener invasive Liriomyza species (Diptera: Agromyzidae) in China"

_PeerJ, doi:10.7717/peerj.10138_

## Round 0.1 · original submission · Major Revisions

Dear Dr. Chang and colleagues:

Thanks for submitting your manuscript to PeerJ. I have now received two independent reviews of your work, and as you will see, the reviewers raised some concerns about the research. Despite this, these reviewers are optimistic about your work and the potential impact it will have on research studying Liriomyza taxonomy and species identification. Thus, I encourage you to revise your manuscript, accordingly, taking into account all of the concerns raised by both reviewers.

Importantly, please address all concerns by the reviewers regarding intraspecific variability. There seems to be concern with the robustness of your molecular analyses.

Please ensure that an English expert has edited your revised manuscript for content and clarity. Please also ensure that your figures and tables contain all of the information that is necessary to support your findings and observations.

Please attempt to better discuss your research and viewpoints within the broader literature on the subjects at hand, and please ensure that critical references raised by the reviewers are included and discussed properly. Please also provide more clarity and transparency with information pertaining to your study design and comparative analyses, especially sampling protocols.

There are many comments by both reviewers that ask for more information on specific issues; please address these.

Please note that Reviewer 2 kindly provided a marked-up version of your manuscript.

I look forward to seeing your revision, and thanks again for submitting your work to PeerJ.

Good luck with your revision,

-joe

·

Basic reporting

The work of Dr. Chang and Collaborators focuses on the possibility to discriminate among three Liriomyza species of agricultural importance. They re-evaluate three morphological characters and develop a new test based on a multiplex PCR.

In my view, the manuscript is not acceptable in its present form, but has the potential to become acceptable following revision.
The one and only issue that, in my view, may prevent it from being acceptable even after revision, if not dealt with convincingly, is intraspecific variability. This is possibly the key reason why such tests are not valid/usable and may mislead other authors into using an invalid test in their research.



Language is not always clear and should be improved before publication. As an example see lines 27-19, 45-49, 57-60, 145-149, 154-157, 192-194.

Background is appropriate, but some recent studies (eg: Chen et al, 2019 Sci.Rep; Carapelli et al, 2018 Genes) may be included. Furthermore it is not always clear form the introduction and discussion sections:
a) which morphological characters are considered as inequivocous and which are considered dubious and re-evaluated. A clear distinction between male genital structures, being inequivocous, and the other three, being equivocous, stems from the results of this study. Does this represent the state of the art before this study? Did other authors comment on the validity of these characters in the past?
b) which are the strength and weaknesses of the molecular tests developed in the past and which is the knowledge gap that the authors are willing to cover. Curiously, the authors suggest that 'using samples from different geographic regions' (line 197) may be a limitation, while I would argue that using samples from one single location is the key limitation of this study (see below).
A paper (Zehnder et al 1986) is referenced to in the text but it is not present in the bibliography. Please cross check text and bibliography throughout.

The structure of the manuscript appears correct.

Figures are generally appropriate.
In Figure 4 please specify what interval the central box represents and add marks (usually asterisks are used) to convey the results of the post-hoc tests.
In Figure 6 please indicate which positions are variable within each species. The image (as is) suggests that sequences are invariable within species, that is certainly not the case.
In Figure 7 it is not clear which species panel B refers to. Please prepare a montage figure where all developmental stages of all species are shown and emend the caption accordingly.
In Table 1 the accession number of the complete mitochondrial genome of the three species may be removed.
Table 2, see below.

No mention is made to data deposition, while I deem that new cox1 sequences should be deposited in GenBank.

Experimental design

The work presented can be considered as original primary research.

The research question is well defined (i.e. developing an improved molecular test for species discrimination) but the knowledge gap this is filling is not well defined or justified. In fact, the limitations of previous molecular methods similarly based on PCR are not presented and it is not clear how this new test improves over available tests.

Some aspects of the methodology are not clearly presented and it cannot be evaluated if the investigation has been conducted under rigorous standards. More specifically, my main concern regards intraspecific variability, an issue that appears to have been overlooked by the authors but, in my view, needs some consideration.
a) how many individuals, and how many from which species, were sequenced (lines 98-102)? Did the authors sequence all individuals listed in Supplementary Table 1?
b) how many sequences were taken from GenBank (lines 103-104)? How many from each species? Which is the geographical representativity of these sequences? Which is the level of intra and interspecific variability?
c) how many variable sites were observed between species (lines 105-107)? How many different sites were fixed and invariable within species?
d) how many specimens were tested to validate the test? Which developmental stages? Three bands per species (adults) and three band from three different developmental stages (stages and species are not indicated) are shown in figure 7. Is this the full panel of tests conducted to validate the essay? I think that a minimum of few tens of individuals per species (including different developmental stages) may be used here.
e) line 153 says that the reverse primer is 'common only to those three Liriomyza species'. I would say that it 'is common' to the three species, while the authors cannot know if it is common 'only' to these. This is relevant as the primer may amplify in other species, something that has not been tested.

Methods (meaning how experiments were performed) are described with sufficient detail. How many samples, and which samples were included is not always clear (see above).

Validity of the findings

Following guidelines of the journal, impact and novelty are not assessed. Nevertheless I think that the applicability of the test should be declared in full clarity.
As from my comments above, I think that natural intraspecies genetic variability is a crucial aspect here. Considering that a) sequence data used for test development may or may not include a sufficient number of reference sequences from outside of China (see my abovestated request for clarification), b) samples used to validate the test are from China (see also my abovestated comment on their number), and c) the fact that we are talking about three invesive species whose native range is not China and may display a sizeable level of genetic diversity in their native range that is not present in China, I would argue that, strictly speaking, the test is valid for China only.
Obviously, the test developed for China by Dr. Chang can be revalidated by other authors working in different areas, but it is important to clearly state the range of applicability of the test as currently proposed.

Underlying data has generally been provided, but a few pieces of information should be addedd:
a) Gen Bank accessions for new sequences (see my comment above).
b) patterns of setae color in Table 2 are not clearly reported. I suggest the authors list, for each species, the visible phenotypes (including uncertain phenotypes that are descibed in the text) and report their absolute and relative frequencies. Here the 'unit of observation' is the individual insect, not the seta. This has been correctly presented in the text but not, in my view, in the table.
c) statistical analysis on wings is correctly conducted (an overall test followed by multiple post hoc tests), but the results are not clearly presented. The p value of the overall test should be reported as well as p values for each of the three post hoc comparisons (or at least the significant ones). Furthermore, in the methods section it is declared that p=0.05 is used as a significance threshold, while in the result section a p=0.001 is considered not significant.

Speculation is not an issue.

Conclusions are generally well stated, apart from the need (presented above) to specify how the morphological reanalysis and the new test improve over current knowledge.

Reviewer 2 ·

Basic reporting

The present manuscript of Dr. Chang and co-Authors describes the efficiency of three morphological characters in the identification of three Liriomyza species (i.e., L. trifolii, L. sativae and L. huidobrensis). Moreover, the Authors developed of a multiplex PCR approach to enhance our ability to identify these three important pest insects.

The English is not always clear and many sentences are vague and imprecise (see attachment for further details). In my opinion, it should be improved, before publication.

The literature references should be improved in some points (e.g., lines 168-170). The background and the aims are sufficiently described and clear throughout the text.

The structure of the manuscript is clear, as well as the figures and tables are relevant to the manuscript aim. However, I would suggest to improve the Figure 1 by both increasing the resolution and by adding more annotation to help the reader understanding the morphological description. I think the Figure references in the text should be improved (see attachment for further details).

The raw data are not shared (e.g., the sequences obtained from their first PCR are not shared and no accession number is given).

Experimental design

The manuscript is an original primary research and the research question is clear and relevant.
However, it does not seem to me that the analyses were conducted rigorously. The way in which the methods and the results are described opens to some important critical issues (also highlighted in the attached pdf). The methods are not sufficiently described to be reproducible. For examples, it is not clear how many sequences they used to design the primers. If the sequences were only one per species, how did the Authors account for intra-specific variability of the cox1 gene? Moreover, no other Liriomyza species was included in the alignment as a control that the primers were actually specific for the three Liriomyza species under study. In the analysis, specimens and sequences of Liriomyza langei were not included and it is known that this latter species cannot be morphologically distinguished from L. huidobrensis (Scheffer et al 2014). Therefore, in my opinion, the Authors should have at least included in the cox1 alignment the sequences of L. langei (available on public databases). In this way, the Authors could confirm that the species-specific selection through multiplex PCR was potentially effective also in other areas where L. huidobrensis is present (e.g., America).
Furthermore, it is not clear from the methods (and the results), whether the Authors sequenced or not the multiplex PCR product to confirm that they actually selected the cox1 of the wanted species.
In my opinion, it is fundamental that these points, just highlighted, are clearly described and satisfied prior further reviewing or publishing the manuscript.

Validity of the findings

In my opinion, the novelty and impact of the present work strongly depends on few aspects. The first one is the inclusion of specimens or at least cox1 sequences of L. langei in the alignment to check the efficiency of the primers designed. The second aspect is the sequencing of the multiplex PCR products to confirm that the primer selection actually works. A third aspect that should be better and clearly described is the application of this method to all the developmental stages of L. sativae and L. huidobrensis. The authors reported that this method worked only for all the developmental stages of L. trifolii. If we cannot distinguish larvae from these three species, then this method is not effective to develop useful control strategies for these pest insects. Finally, the submission of the cox1 raw data on public databases is fundamental for reproducibility.
Without these improvements, the present work is inconclusive and ambiguous, with the results not fully supported in the present form.

Annotated reviews are not available for download in order to protect the identity of reviewers who chose to remain anonymous.

---

## Round 0.2 · accepted · Accept

Dear Dr. Chang and colleagues:

Thanks for once again resubmitting your manuscript to PeerJ. I now believe that your manuscript is suitable for publication. Congratulations! I look forward to seeing this work in print, and I anticipate it being an important resource for groups studying Liriomyza taxonomy and species identification. Thanks again for choosing PeerJ to publish such important work.

Best,

-joe

·

Basic reporting

The current version of the ms marks a significant improvement over the first submission, especially in terms of language and clarity in the presentation of the results.
Some of the limitations - i.e. geographical representativeness and reliance on few sequences - are still present, but giving a clear account of these aspects in the manuscript now allows the reader to decide if and when to apply the test proposed.

Experimental design

NA

Validity of the findings

NA

Additional comments

NA